# Enhancement of osmotic stress tolerance in soybean seed germination by bacterial bioactive extracts

**Sang Tae Kim**[1,2]**, Mee Kyung Sang**[1]*

**1** Division of Agricultural Microbiology, National Institute of Agricultural Sciences, Rural Development Administration, Wanju, Republic of Korea, **2** Department of Applied Bioscience, Dong-A University, Busan, Republic of Korea

* mksang@korea.kr

**Data Availability Statement:** All dataset are available from the Figshare database (http://doi.org/10.6084/m9.figshare.24041850).

## Abstract

Soybean (*Glycine max* (L.) Merr.) is important to the global food industry; however, its productivity is affected by abiotic stresses such as osmosis, flooding, heat, and cold. Here, we evaluated the bioactive extracts of two biostimulant bacterial strains, *Bacillus butanolivorans* KJ40 and *B. siamensis* H30-3, for their ability to convey tolerance to osmotic stress in soybean seeds during germination. Soybean seeds were dip-treated in extracts of KJ40 (KJ40E) or H30-3 (H30-3E) and incubated with either 0% or 20% polyethylene glycol 6000 (PEG), simulating drought-induced osmotic stress. We measured malondialdehyde content as a marker for lipid peroxidation, as well as the activity of antioxidant enzymes, including catalase, glutathione peroxidase, and glutathione reductase, together with changes in sugars content. We also monitored the expression of genes involved in the gibberellic acid (GA)-biosynthesis pathway, and abscisic acid (ABA) signaling. Following osmotic stress in the extract-treated seeds, malondialdehyde content decreased, while antioxidant enzyme activity increased. Similarly, the expression of GA-synthesis genes, including *GmGA2ox1* and *GmGA3* were upregulated in KJ40E-dipped seeds at 12 or 6 h after treatment, respectively. The ABA signaling genes *GmABI4* and *GmDREB1* were upregulated in H30-3E- and KJ40E-treated seeds at 0 and 12 h after treatment under osmotic stress; however, *GmABI5*, *GmABI4*, and *GmDREB1* levels were also elevated in the dip-treated seeds in baseline conditions. The GA/ABA ratio increased only in KJ40E-treated seeds undergoing osmotic stress, while glucose content significantly decreased in H30-3E-treated seeds at 24 h after treatment. Collectively, our findings indicated that dip-treatment of soybean seeds in KJ40E and H30-3E can enhance the seeds' resistance to osmotic stress during germination, and ameliorate cellular damage caused by secondary oxidative stress. This seed treatment can be used agriculturally to promote germination under drought stress and lead to increase crop yield and quality.

**Funding:** This work was supported by National Institute of Agricultural Sciences (Project no. PJ01587101) of Rural Development Administration, Republic of Korea. The funders had no role in study design, data collection and analysis, decision to publish, or preparation of the manuscript.

# Introduction

Soybeans (*Glycine max* (L.) Merr.) are major crops owing to their versatility, nutritional value, and use in several industrial applications, and are grown in various countries around the world [1, 2]. Soybean cultivation depends on region and crop variety, and generally begins in the spring between early April and late May, with a maturation stage lasting from 100 to 150 days after sowing, and a harvest season during fall [3]. The production of soybeans can be affected by abiotic factors, including drought, salt, and flooding, as well as biotic factors, such as plant pathogens and pests, that collectively determine the final crop yield and its quality [4, 5]. Therefore, to prevent infection by plant pathogens and insects after planting, soybeans require regular disease and insect management, which can be achieved through a variety of fungicides and pesticides, mechanical cultivation, the use of cover crops, and disease-resistant cultivars [6, 7]. Moreover, previous studies have investigated physiological, biochemical, and molecular aspects [8–11] in an attempt to comprehend mechanisms of abiotic stress tolerance and identify potential applications in soybean [12, 13].

Soybeans are highly sensitive to water stress, and it can severely impact their growth and production. Drought damage can occur at several stages of soybean development, especially during the early stages of growth, which can lead to a decrease in germination rate and sprout emergence, as well as growth delay or stunting [14, 15]. This can result in smaller and less vigorous plants with fewer branches and leaves [16, 17]. Seed germination is influenced by environmental conditions such as water, light, and temperature, though moisture has a significant impact on seed germination, which is divided into three phases (phase I: imbibition, phase II: lag phase, phase III: radical emergence) [18, 19]. Phytohormones also play a role, including abscisic acid, which is related to seed dormancy, and gibberellic acid (GA), which is involved in seed germination and development [19]. For example, the overexpression of ABA response genes, *ABI4* and *ABI5*, showed minimal germination than wild-type, and in the case of the *GA2ox* mutant, which deactivate bioactive GA, a reduction in seed dormancy was observed [20–24]. During the plant's maturation, drought stress also causes wilting, leaf yellowing, premature leaf drop, and accumulated abscisic acid (ABA) owing to increased water loss through transpiration, stomatal conductance in response to elevated temperatures [25]. If drought stress persists, this can negatively impact flower and pod production as well as seed quality, which may be seen in the reduced seed size and weight, and the dropping of premature pods [26, 27]. To alleviate drought stress, a range of soybean crop management strategies have been developed, including planting drought-tolerant cultivars, managing irrigation, and employing alternatives such as trehalose, and biostimulants using microorganisms [12, 13, 28–30].

Biostimulants refer to microorganisms or microorganism-derived compounds, that can enhance plant growth, development, and stress tolerance [31]. These can be applied to alleviate drought stress by increasing the ability of plants to absorb and retain water [32]. Biostimulants can improve the soil structure and increase plant root development, allowing plants to absorb water more efficiently and increase their water-holding capacity, in addition to enhancing plant antioxidant and osmotic adjustment capacities, which together allow for the development of a level of tolerance to drought-induced stress [33, 34]. Drought stress may cause secondary oxidative stress in plant cells, leading to cell death and reduced growth [35]. In this context, biostimulants can increase the production of antioxidants and protect plants from oxidative damage [36]. Moreover, they facilitate plant adjustment to osmotic stress by upregulating osmoprotectants production, such as proline, which can help toward maintaining water balance and prevent cell damage [37]. Biostimulants may finally improve plant nutrient uptake and metabolism, which can enhance stress resistance and enable the more efficient utilization of available nutrients [38]. Together, these qualities make biostimulants attractive tools for

conferring osmotic stress tolerance to crops, and thereby mitigate the more adverse effects of drought in plants grown in drought-prone environments.

In our previous study, we identified three bacterial strains, *Bacillus aryabhattai* H26-2, *B. siamensis* H30-3, and *Peribacillus butanolivorans* KJ40, as biostimulant candidates for alleviating plant abiotic stresses, including heat and drought [39, 40]. Strains H26-2 and H30-3 could promote plant growth, decrease leaf wilting, and enhance recovery after rewatering against heat and drought stress in Chinese cabbage, by regulating leaf abscisic acid (ABA) content and stomatal opening [39]. In particular, strain H30-3 produces exopolysaccharides that play a pivotal role in the mitigation of heat and drought stress in plants [39]. Strain KJ40 modulates antioxidant activities, such as peroxidase and glutathione peroxidase, allocated polyphenol contents, including flavonoids, of pepper plants that were impacted by drought, and could influence the development of stress tolerance in plants [40].

Recently, soybean cultivation and production in Korea, has increased from 50,638 ha cultivation area and 177 kg production per 10 a in 2018, to 63,956 ha and 203 kg per 10 a in 2022 [41], however, cumulative precipitation during the six months from December 2021 to May 2022 was 167.9 mm, which is 49.1% of the standard normal precipitation from 1991 to 2020, soybean crops could have been stressed by a lack of rainfall after sowing in the field. Given the importance of a robust stress tolerance against drought, the objectives of this study were: (1) to identify bacterial bioactive extracts resistant to osmotic stress, in soybean seeds, that could maintain efficient germination under stress-inducing conditions, and (2) to explore seed physiological changes, including those related to lipid peroxidation, antioxidant activities, osmotic and germination-related hormone gene expression, and reducing sugar content in the selected bioactive extracts, under osmotic stress conditions during germination.

## Materials and methods

### Extract preparation of drought-tolerant inducing bacteria

*Bacillus butanolivorans* KJ40, *B. siamensis* H30-3, and *B. aryabhathai* H26-2 were used as drought tolerance-inducing bacteria as described in our previous studies [39, 40]. The strains were cultured in 1 L of tryptic soy broth (TSB, Difco, USA) at 28°C, for three days, and then the cultures were centrifuged at $6,000 \times g$ for 30 min. The supernatants were collected and filtered through a 0.45 μm vacuum filter. The filtrates were serially fractionated with organic solvents (1:1, v/v) including hexane, dichloromethane, ethyl acetate, and n-butanol, according to polarity, by using a funnel on a shaker at 120 rpm overnight. Each separated extract was vacuum-evaporation (KJ40 extract was hexane: 110 mg, dichloromethane: 80 mg, ethyl acetate: 93 mg, n-butanol: 320 mg; H30-3 was hexane: 50 mg, dichloromethane 130 mg, ethyl acetate: 73 mg, n-butanol: 230 mg; H26-2 was hexane: 90 mg, dichloromethane: 110 mg, ethyl acetate: 180 mg, n-butanol: 220 mg) and dissolved in methanol. The concentrated extracts were diluted in water.

### Screening for drought tolerance -inducing bacterial extracts on soybean seeds

*Glycine max* 'Daewon' was surface-sterilized in 2% sodium hypochlorite, and then soaked in different concentrations of bacterial extracts (1, 10, and 100 μg/mL) for 1 h at 25°C and 100 rpm, before blotting onto sterile filter papers to remove excess solution. To examine the drought-tolerance-inducing activity of the extracts, 20 mL of 20% polyethylene glycol (PEG) 6000 (81260, Sigma, USA) (-0.49MPa) [42] with a half-diluted Hoagland solution was placed in a Petri dish (diameter 90 mm) with two-layered filter papers (No. 2, Whatman, England),

and soaking-treated seeds were placed on a plate. Ten soybean seeds were used for selection of bioactive fractions, and then 25 seeds were tested for determination of the effective concentration of each fraction. Seven days after incubation at 25°C and 16 h light/8 h dark, we evaluated the final germination percentage (FGP, %), median germination time (T50, day), and mean germination time (MGT, day). The calculations were as follows: FGP (%) = final day of germination seeds/total seeds × 100; T50 = ti + [(N/2-ni) (ti-tj)]/ni-nj (where N = final number of germinated seeds; ni, nj = accumulative number of seeds germinated by adjacent counts at times ti and tj, respectively [ni < N/2 < nj]); MGT = Σ(nxd)/N, n = number of germinated seeds on each day (where d = number of days from the beginning of the experiment; and N = total number of germinated seeds at the end of the test) [43].

## Malondialdehyde, glutathione, and antioxidant enzyme activity

Seeds were sampled at 0, 3, 6, 9, 12, 24, 48, and 72 h after PEG treatment. For the seed physiological assay, three seeds were pooled and grounded using a mortar and pestle, prior to collection. The malondialdehyde (MDA) content was determined as described by Dhindsa et al. [44]. Ground seeds (100 mg) were homogenized in 500 μL of 0.1% trichloroacetic acid (TCA) (w/v) and centrifuged for 10 min at 13,000 × $g$ at 4°C. The supernatant (200 μL) was mixed with 600 μL of 20% TCA amended with 0.5% 2-thiobarbituric acid (TBA); incubated at 90°C for 30 min. The reaction was inhibited on ice for 5 min., and the absorbance at 450, 532, and 600 nm were measured by a spectrophotometer (Infinite M200 PRO, TECAN, Switzerland). The MDA content was calculated according to Bao et al. [45]. Glutathione (GSH, 703002; Cayman, USA) and antioxidant enzyme activities, including those of superoxide dismutase (SOD, 706002; Cayman, USA), catalase (CAT, 707002; Cayman, USA), glutathione peroxidase (GPX, 703102; Cayman, USA), and glutathione reductase (GR, 703202; Cayman, USA), were detected according to the manufacturer's instructions.

## Seed RNA extraction and qRT-PCR

Seeds were sampled and grounded as already described. Total RNA from the seeds was extracted using the TRIzol™ reagent (Invitrogen, USA), and RNA concentration in our samples was measured in Nanodrop (ND-1000, Thermo Scientific, USA). Total RNA was synthesized into cDNA using TOPscript™ RT DryMIX (dT18) (Cat. No. RT200; Enzynomics) and diluted to 1:10. For quantitative real-time (qRT) PCR, we used a CFX96 Real-time PCR Detection system (Bio-Rad), together with specific primers for the studied genes (Table 1). The qRT-PCR was performed as follows: 95°C for 10 min (initial denaturation); 45 cycles of 95°C for 15 s (denaturation), 58°C for 60 s (annealing), and 72°C for 45 s (extension). To normalize *Glycine max* gene expression, *GAActin11* was used as a reference gene, and the relative expression level was determined using the $2^{-\triangle\triangle Ct}$ method [50].

## Abscisic acid, gibberellic acid, and sugars contents

Abscisic acid content was measured as described by Lim et al [51]. Crushed seed (0.1 g) was extracted in 2ml of methanol containing 500 mg/L citric acid monohydrate and 100 mg/L butylated hydroxyl toluene, at 4°C overnight on a rotary shaker. The mixture was centrifuged at 2000 × $g$ for 10 min, and the supernatant was transferred and dried using a speed vac (CVE-2000, EYELA). Samples were quantified using a Phytodetek-ABA kit (Agdia Inc., USA) according to the manufacturer's instructions. Gibberellic acid (GA, MBS9310617, MyBioSource) and sugars (K-SUFRG, Megazyme, Ireland), including sucrose, D-fructose, D-glucose, were evaluated according to the manufacturer's protocol.

**Table 1. Primers of soybean genes used in the real-time PCR analysis.**

| Primer name | | Sequence (5'-3') | Description | Reference |
|---|---|---|---|---|
| GmGA20ox1 | F | CAAAACACATCCGCAGAAA | Gibberellin 20-oxidase1 | [46] |
| | R | GTTGGACTGGTTTGGCATCA | | |
| GmGA3ox1 | F | GCGTAACGGTGAATAGGACAT | Gibberellin 3-oxidase 1 | [46] |
| | R | CGAGCAACAGAGTGAACCAA | | |
| GmGA2ox1 | F | ACATTGGCTTTGGAGAGCAT | Gibberellin 2-oxidase 1 | [46] |
| | R | GGAAATCTGAAGGCCATCAA | | |
| GmGA3 | F | AGATTGAACGCACCACACCTT | *ent*-kaurene oxidase | [47] |
| | R | TCGCAGGAAGAAGAAGAGGATAGA | | |
| GmABI5 | F | CGAGTTCCAGCACAGTCT | Abscisic acid-insensitive 5 | [48] |
| | R | TGTTCTCTTCAGCGTTCCA | | |
| GmABI4 | F | GAATCAACAGCAACAGCAACA | Abscisic acid-insensitive 4 | [48] |
| | R | ACCGAAGAAGCATCCATAGC | | |
| GmDREB1 | F | GTAAAGATTGTTCGTATGGGACAAG | Dehydration-responsive element binding protein 1 | [49] |
| | R | ACACCTAAAATGAGCAACCGTACTA | | |
| Actin11 | F | ATCTTGACTGAGCGTGGTTATTCC | Actin11 | [48] |
| | R | GCTGGTCCTGGCTGTCTCC | | |

## Statistical analysis

All data were analyzed using the Statistical Analysis System (SAS) (version 9.4, SAS Institute Inc., Cary, NC, USA). Fraction and concentration selections were conducted with six and eight replications (fraction selection: 10 seeds/replication, concentration selection: 25 seeds/replication) from two experiments, while MDA, antioxidant enzyme activities, and sugars (sucrose, glucose, and fructose) analyses were conducted with nine replicates, obtained from three experiments. Gene expression was performed with six replicates from two experiments. Each replicate was randomly combined with three seeds. Data from repeated experiments were pooled after confirming the homogeneity of variance using Levene's test. After the ANOVA, the least significant difference (LSD) or Tukey's test was conducted for statistical comparisons between groups.

## Results

### Selection of bacterial bioactive extracts promoting seed germination under osmotic stress conditions

Final seed germination was reduced under drought stress conditions (KJ40; $F$ value = 27.01, $P$ <0.0001, H30-3; $F$ value = 39.56, $P$ <0.0001; H26-2; $F$ value = 39.49, $P$ <0.0001) (S1 Table). The seed dipping treatment in selected bacterial extracts (100 μg/ml) promoted the development of resistance to drought-induced stress and ameliorated the decrease of seed germination (KJ40; $F$ value =2.43, $P$ = 0.0767, H30-3; $F$ value =9.26, $P$ = 0.0005, and H26-2, $F$ value =0.69, $P$ = 0.6044) (S1 Fig). The strain KJ40 and H30-3-extracts by ethyl acetate significantly increased soybean seed FGP compared to control under PEG 20% conditions (S1 Fig). By contrast, the effect of strain H26-2 extract was less pronounced and not significantly different compared to the control, as such it was excluded from further study in this work (S1 Fig).

After dipping treatment of various concentration (1, 10, 100 μg/mL) of the extracts by ethyl acetate of strains KJ40 and H30-3, we observed that treatment had a significant effect on FGP ($F$ value = 6.18, $P$ <0.0001). The 100 μg/mL of H30-3 and 1 μg/mL of KJ40 extracts effectively increased FGP compared to control under PEG 20% conditions. The FGP of seeds dipped-

**Table 2. Soybean seed final germination percentage, median germination time, and mean germination time.**

| Treatment | Concentration (μg/mL)[1] | Final germination percentage (FGP, %) | Median germination time (T50, day) | Mean germination time (MGT, day) |
|---|---|---|---|---|
| Control | 0 | 57.46 ± 5.56 d [2] | 4.00 ± 0.32 a | 4.67 ± 0.26 a |
| H30-3 | 1 | 74.00 ± 3.30 bc | 3.83 ± 0.13 a | 4.35 ± 0.10 a |
| | 10 | 75.20 ± 1.00 bc | 4.01 ± 0.18 a | 4.52 ± 0.17 a |
| | 100 | 83.11 ± 3.51 a | 3.95 ± 0.23 a | 4.51 ± 0.18 a |
| KJ40 | 1 | 79.56 ± 3.30 ab | 4.10 ± 0.21 a | 4.53 ± 0.18 a |
| | 10 | 69.33 ± 2.31 c | 4.29 ± 0.22 a | 4.73 ± 0.19 a |
| | 100 | 71.50 ± 1.92 bc | 4.02 ± 0.17 a | 4.56 ± 0.16 a |

[1] The concentration of the bacterial supernatants extracted by ethyl acetate.

[2] Small latter means significant difference by the LSD test at $P < 0.05$

treated with 100 μg/mL of H30-3 and 1 μg/mL of KJ40 extracts were 89.3 ± 2.2%, and 85.3 ± 1.3, respectively; that of control was 64.5 ± 5.5 (Table 2). However, T50 and MGT values were similar to those of the control group (Table 2). Therefore, 100 μg/mL of H30-3 extract (H30-3E) and 1 μg/mL of KJ40 extract (KJ40E), isolated in ethyl acetate, were selected for further study.

## Changes in seed MDA and antioxidant enzyme activities during germination under osmotic stress

Osmotic stress caused by PEG 20% treatment during seed germination triggered secondary oxidative stress, as indicated by the increased production of the oxidative stress marker MDA, compared to the control PEG 0% treatment (3 h after treatment; $F$ value = 10.52, $P$ value = 0.0022, 6 h after treatment; $F$ value = 1.59, $P$ value = 0.2137, 9 h after treatment; $F$ value = 4.76, $P$ value = 0.0342, 12 h after treatment; $F$ value = 0.03, $P$ value = 0.872, 18 h after treatment; $F$ value = 5.61, $P$ value = 0.0221, 24 h after treatment; $F$ value = 5.83, $P$ value = 0.02, 48 h after treatment; $F$ value = 2.74, $P$ value = 0.1045, 72 h after treatment; $F$ value = 6.36, $P$ value = 0.0151) (Fig 1). Under non-stressed conditions, MDA content in soybean seeds was not affected by dipping treatments with H30E and KJ40E, except at 9 and 48 h after treatment (Fig 1A). However, H30-3E and KJ40E significantly reduced the MDA content compared to the control, from 12 to 72h after treatment under osmotic stress conditions (Fig 1B).

To scavenge reactive oxidative stress at the early germination stage, we measured several antioxidant enzyme activities, including catalase at 3 and 6 h after treatment, and glutathione peroxidase at 12 h after treatment for H30-3E-dipped seeds. For KJ40E-dipped seeds, we examined activity of catalase at 6 and 12 h after treatment, glutathione peroxidase at 9 h after treatment, and glutathione reductase at 12 h after treatment. All measured antioxidant enzyme activities were significantly increased in the treatment groups compared to controls (Fig 2B). However, superoxide dismutase activity did not differ between the treatments, regardless of PEG conditions (Fig 2).

## Gibberellin and ABA-related genes expression

Gibberellic acid (GA) biosynthesis and ABA signaling genes were evaluated using qRT-PCR. Under non-stressed conditions, relative expression of GA biosynthesis-related genes in H30-3E-dipped seeds, including *GmGA20ox1* and *GmGA3ox1* levels at 12 h after treatment, and *GmGA2ox1* at 0, 6, and 12 h after treatment, were upregulated (At 12 h after treatment: *GmGA20ox1*; 7.37-fold increase, *GmGA3ox1;* 8.53-fold increase, *GmGA2ox1* increase in

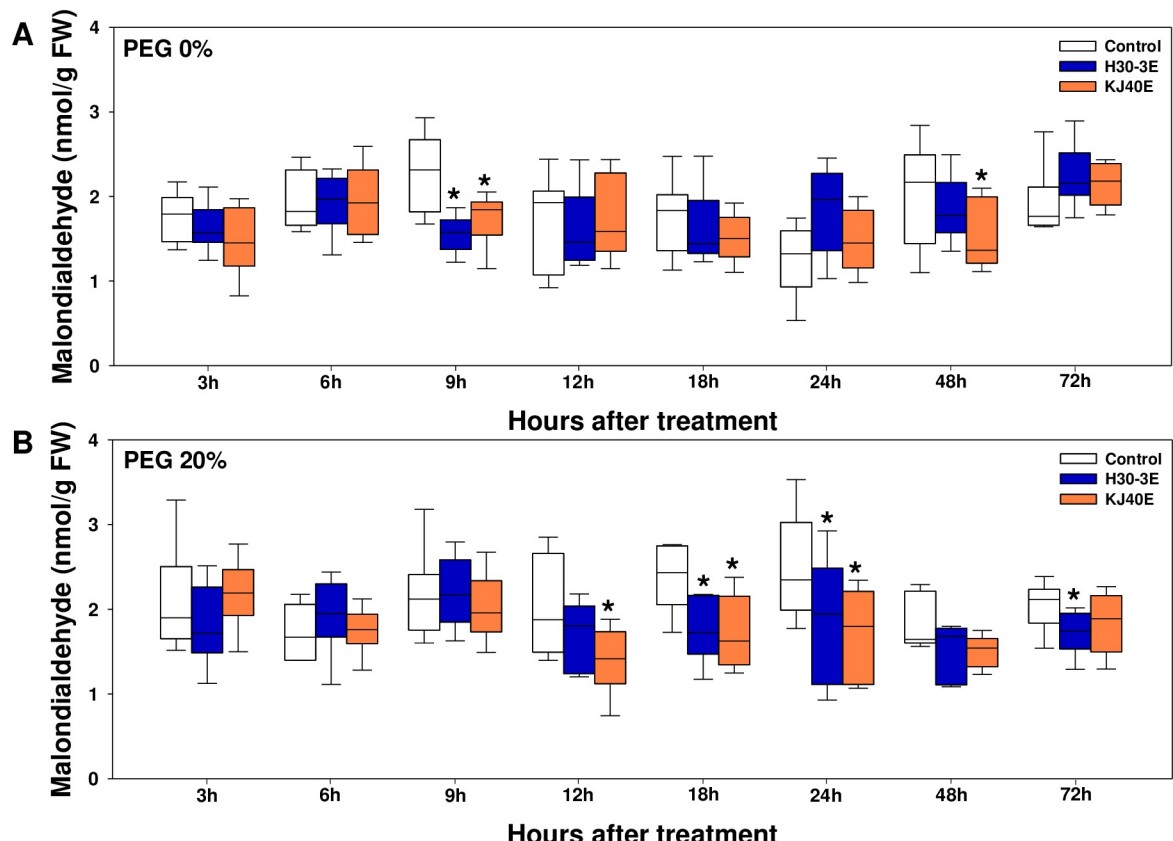

**Fig 1. Changes in malondialdehyde content in soybean seeds dip-treated with control, H30-3 extract (H30-3E), and KJ40 extract (KJ40E) under 0 and 20% of PEG conditions.** The MDA content was measured at 3, 6, 9, 12, 18, 24, 48, and 72 hours after PEG-treatment (HAT). (A) 0% PEG; (B) 20% PEG. Data presented as means ± standard error (n=9, statistical significance assessed by the LSD test, $^*P < 0.05$).

expression at 0 h after treatment; 1.52-fold, at 6 h after treatment; 3.27-fold, and 12 h after treatment; 13.13-fold, all relative to control). However, expression of *GmGA3* at 6 h after treatment decreased (0.55-fold). In contrast, in KJ40E-dipped seeds, relative expression levels of GA synthesizing genes, including *GmGA20ox1* and *GmGA3ox1* at 0h after treatment, and *GmGA3* at 0 and 6 h after treatment, were downregulated compared to control (*GmGA20ox1*; 0.35-fold decrease, *GmGA3ox1*; 0.46-fold decrease, and *GmGA3*-0 h after treatment; 0.43-fold decrease, *GmGA3*—6 h after treatment; 0.48-fold decrease). However, we observed increased expression for *GmGA3ox1* (9.30-fold), *GmGA2ox1* (15.16-fold), and *GmGA3* (6.34-fold) at 12 h after treatment, in relation to the control (Fig 3A). Under PEG 20%-induced osmotic stress conditions, the relative expression of GA biosynthesis-related genes was not significantly affected by H30-3E-dipping treatment, except for *GmGA3ox1* at 6 h after treatment and *GmGA2ox1* at 0 h after treatment (*GmGA3ox1*; 0.40-fold decrease, and *GmGA2ox1*; 2.09-fold increase, relative to the corresponding control) (Fig 3B). In the case of KJ40E-dipped seeds, GA biosynthetic genes were downregulated at 0 h after treatment relative to controls (*GmGA20ox1*; 0.29-fold, *GmGA3ox1*; 0.32-fold, *GmGA2ox1*; 0.71-fold, and *GmGA3*; 0.49-fold); however, the expression of *GmGA2ox1* at 12 h after treatment and *GmGA3* at 6 h after treatment was significantly higher than that of the reference group (*GmGA2ox1*; 2.49-fold and *GmGA3*; 2.04-fold) (Fig 3B).

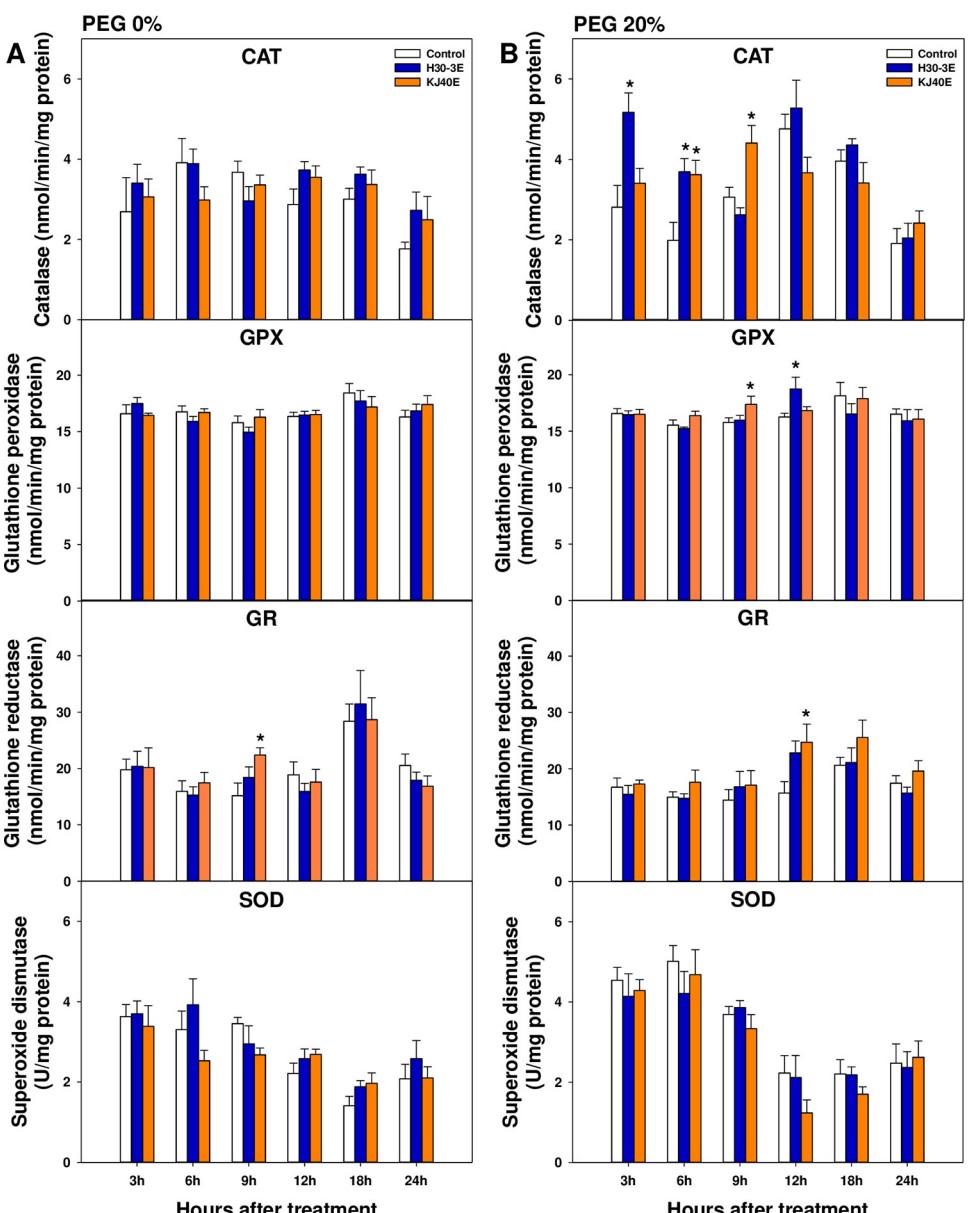

**Fig 2. Antioxidant enzyme activities, including catalase, glutathione peroxidase, glutathione reductase, and superoxide dismutase, in soybean seeds dip-treated with control, H30-3 extract (H30-3E), and KJ40 extract (KJ40E), and incubated in 0 or 20% PEG.** The enzyme activities were evaluated at 3, 6, 9, 12, 18, and 24 hours after PEG-treatment (HAT). (A) 0% PEG; (B) 20% PEG. Data presented as means ± standard error (n=9, statistical significance assessed by the LSD test, *$P < 0.05$).

The relative expression of ABA-signaling genes transiently increased at 12 h after treatment in H30-3E- and KJ40E-dipped seeds. Under baseline conditions, we observed an increase in levels of *GmABI5* at 0 (2.66-fold) and 12 h after treatment (9.11-fold) in H30-3E-dipped seeds, and of *GmDREB1* at 12 h after treatment (3.93-fold) in KJ40-dipped seeds (Fig 4A). Following osmotic stress induction, in H30-3E-dipped seeds the *GmABI4* and *GmDREB1* levels at 12 h after treatment were significantly upregulated (*GmABI4*; 5.19-fold and *GmDREB1*; 2.43-fold), as were the levels of *GmABI4* at 12 h after treatment (4.41-fold) in KJ40E-dipped seeds, in

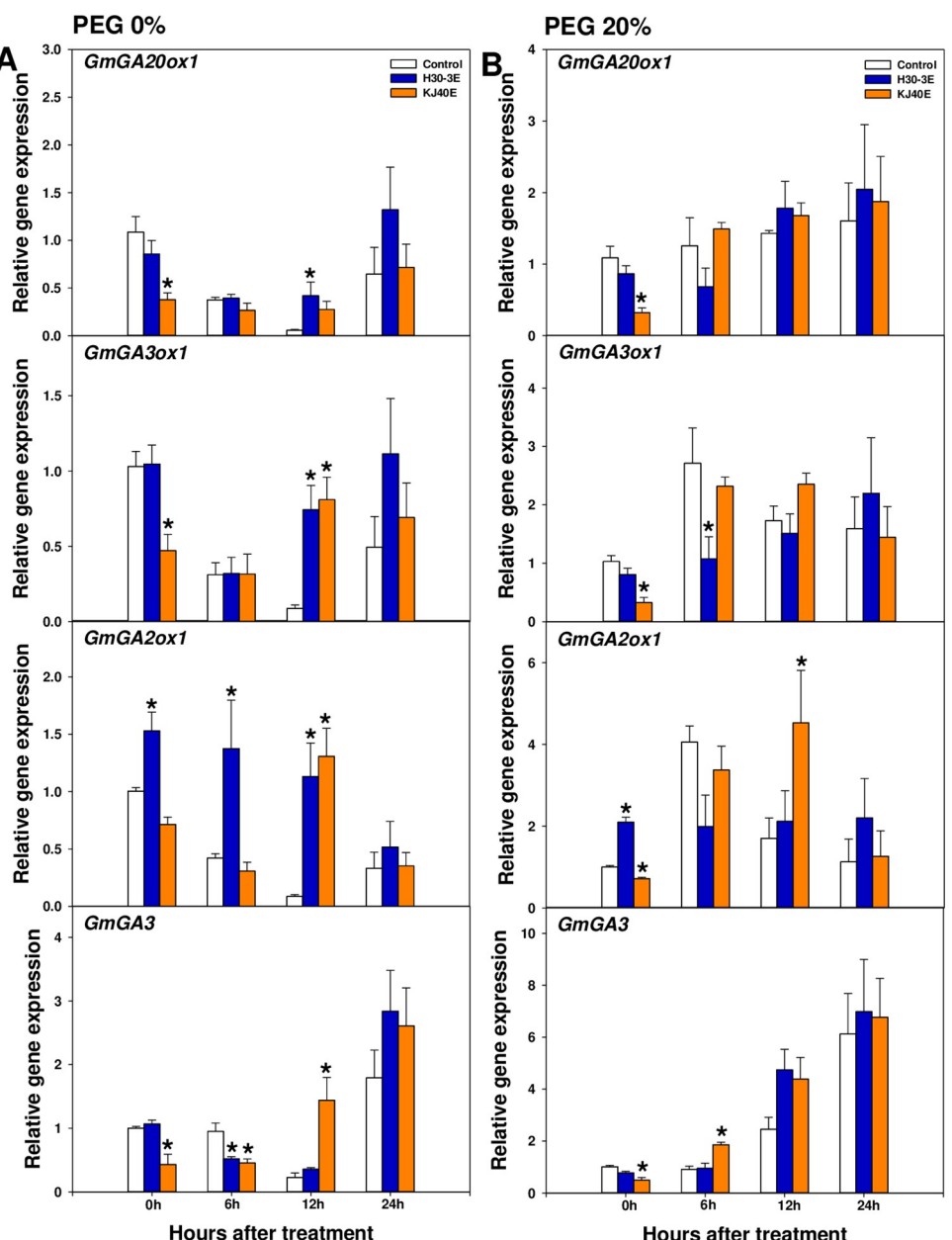

**Fig 3. The relative expression of gibberellin-synthesis genes, including *GmGA20ox1*, *GmGA3ox1*, *GmGA2ox1*, and *GmGa3*, in soybean seeds dip-treated with control, H30-3 extract (H30-3E), and KJ40 extract (KJ40E), and incubated with 0 or 20% PEG.** Gene expression was measured at 0, 6, 12, and 24 hours after PEG-treatment (HAT). (A) 0% PEG; (B) 20% PEG. Data presented as means ± standard error (n=6, statistical significance assessed by the Tukey test, *$P < 0.05$).

comparison to the control (Fig 4B). Evaluation of GA and ABA content in soybean seeds at 24 h after treatment showed that ABA content was not significantly different between baseline and osmotic stress conditions; however, the GA content and GA/ABA ratios in KJ40E-dipped seeds were higher than those in the control following osmotic stress (GA; $F = 5.69$, $P = 0.0098$, GA/ABA; $F = 4.22$, $P = 0.103$) (Fig 5).

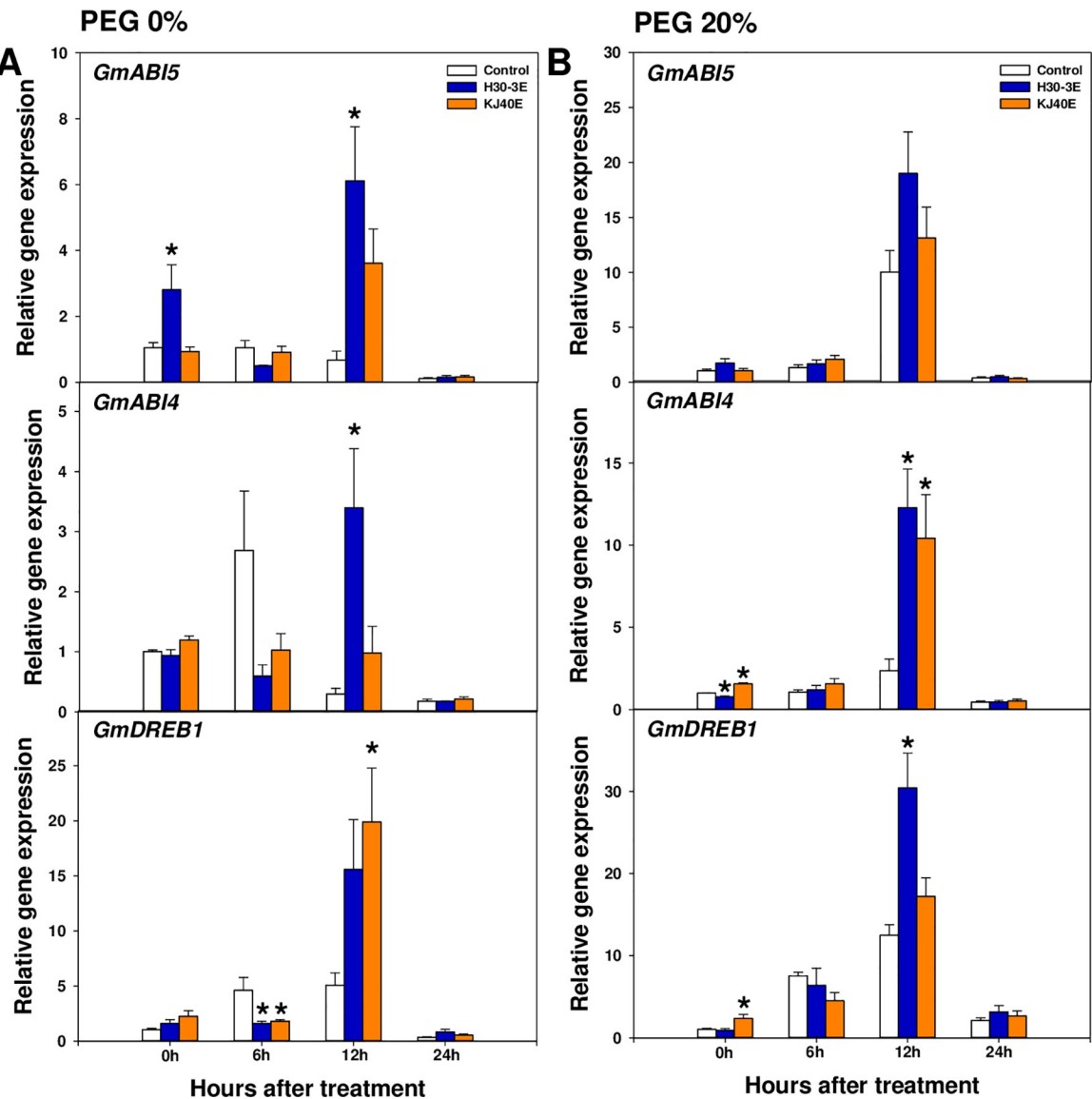

**Fig 4. The relative expression of abscisic acid-related genes, including *GmABI5*, *GmABI4*, and *GmDREB1* in soybean seeds dip-treated with control, H30-3 extract (H30-3E), and KJ40 extract (KJ40E), and incubated with 0 or 20% PEG.** Gene expression was measured at 0, 6, 12, and 24 hours after PEG-treatment (HAT). (A) 0% PEG; (B) 20% PEG. Data presented as means ± standard error (n=6, statistical significance assessed by the Tukey test, *$P < 0.05$).

## Changes in sucrose, glucose, and fructose contents

Glucose levels significantly decreased at 24 h after treatment in H30-3E-treated seeds under drought-simulating stress ($F = 2.77$, $P = 0.0868$), however, sucrose and fructose levels did not significantly differ between conditions. Sugars showed a decreasing pattern, but glucose showed a trend for increase in the control under osmotic stress conditions (Fig 6).

## Discussion

In our previous studies, two bacterial strains, *Bacillus siamensis* H30-3 and *B. butanolivorans* KJ40, were shown to ameliorate the more adverse effects related to heat and drought stress in

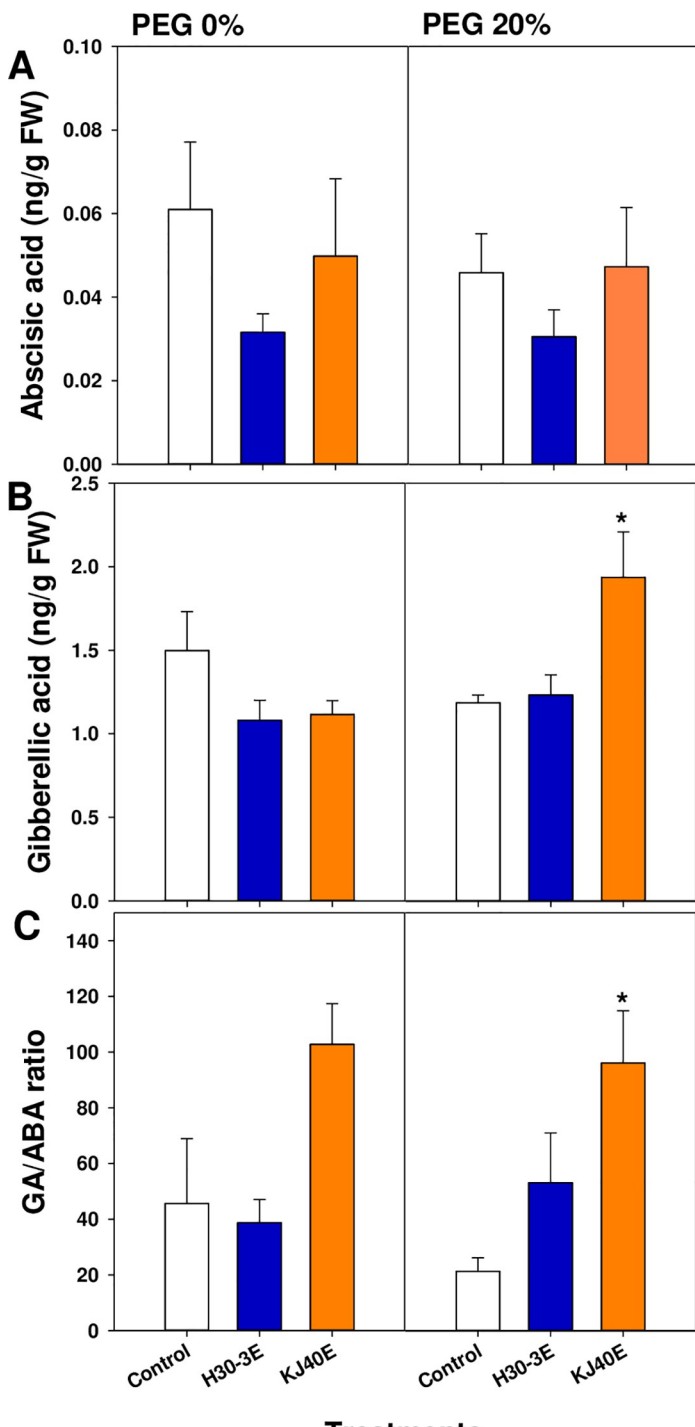

**Fig 5.** Contents of abscisic acid (A) and gibberellic acid (B), and the GA/ABA ratio (C) in soybean seeds at 24 hours after PEG-treatment (HAT). Left column, 0% PEG; Right column, 20% PEG. Data presented as means ± standard error (n=9, statistical significance assessed by the LSD test, *$P < 0.05$).

Chinese cabbage, as well as drought stress in pepper plants [39, 40]. These mitigative activities result from indirect effects on the plants through tolerance development, including the regulation of stomatal opening and production of antioxidant and phenolic compounds. To more

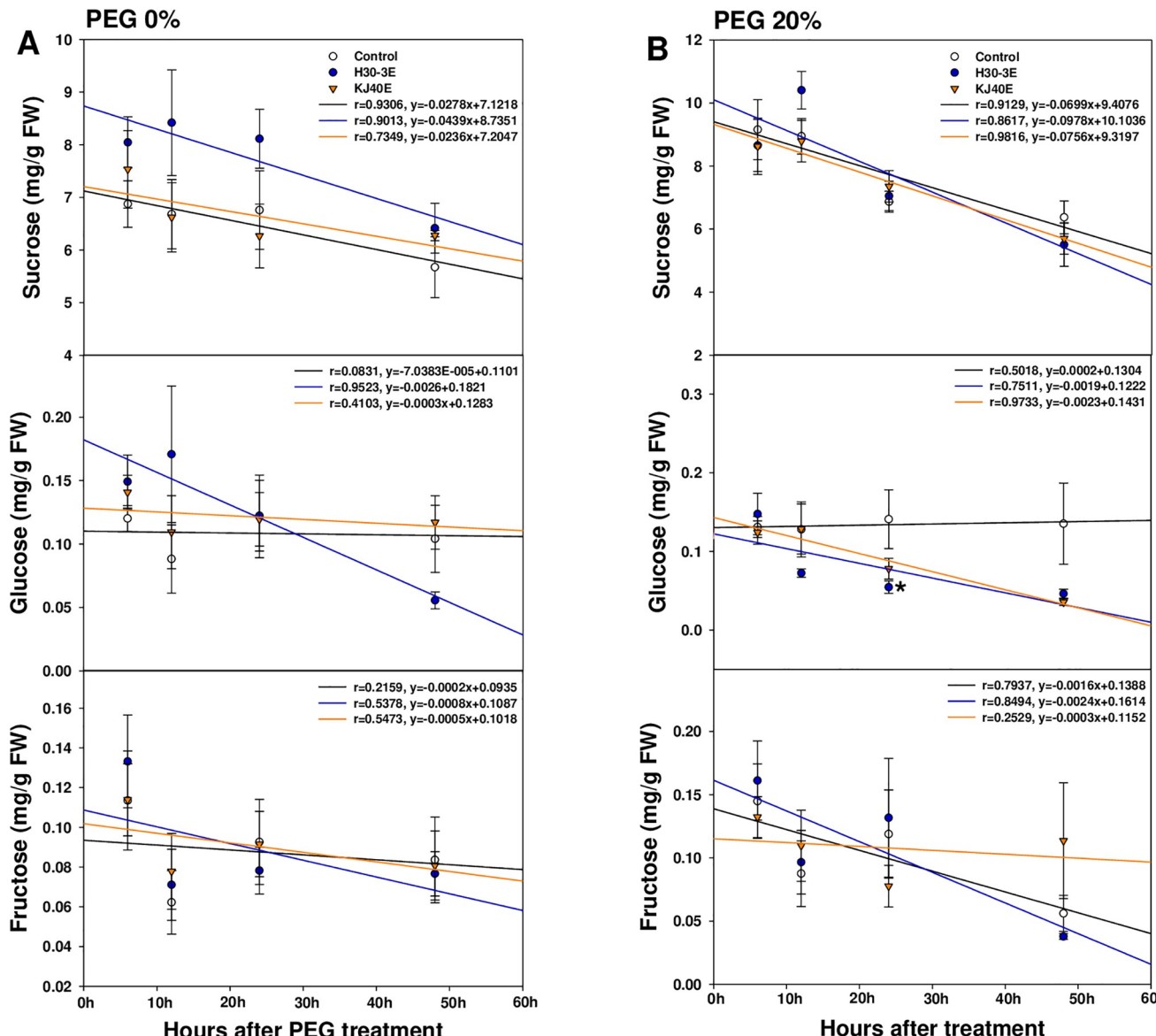

**Fig 6. Changes in the contents of sucrose, glucose, and fructose, in soybean seeds during germination.** (A) 0% PEG; (B) 20% PEG. Data presented as means ± standard error (n=9, statistical significance assessed by the LSD test, *$P < 0.05$).

efficiently protect plants from being affected by unfavorable environmental conditions, alternative application strategies related to plant cultivation must be explored. As such, here we investigated whether the bacterial extract of H30-3 and KJ40 could decrease the impact of osmotic stress on soybean seeds during the germination stage, and indirectly promote the development of resistance to osmotic stress through induction of physiological changes in seeds. Our results demonstrated that the two bacterial extracts alleviated osmotic stress in soybean seeds and increased germination via different mechanisms.

Seed germination is the most important and sensitive stage in the plant life cycle. However, osmotic stress can cause secondary oxidative stress, low germination rates, and delayed germination times [52, 53]. In seeds dip-treated with the extracts of H30-3 at 100 μg/mL and KJ40 at 1 μg/mL, the measured FGP was higher than that of the un-treated control; however, MGT

and T50 did not differ between treatments. This implies that the H30-3 and KJ40 extracts did not affect the rate of germination under osmotic stress conditions induced by 20% PEG treatment. To investigate the pathways related to the observed increase in FGP in the bacterial extract-treated seeds, we measured MDA levels as a readout for assessing secondary oxidative stress and lipid peroxidation state, as well as examined antioxidant enzyme activity. The MDA is used as an oxidative stress indicator, and the germination percentage increases with a decrease in oxidative stress. Our findings indicated that MDA content was higher in seeds subjected to osmotic stress relative to the reference group. Goharrizi et al. [54] have reported that the MDA content positively correlates with PEG concentration. In our study, MDA generally decreased in H30-3E- or KJ40E-dipped seeds from 12 hours after PEG treatment, suggesting that the two extracts could contribute to the mitigation of secondary oxidative stress effects during germination of soybean seeds under osmotic stress conditions. In addition, in the early stages of germination, catalase, and glutathione peroxidase activities were significantly higher in the two extract-dipped seed groups than in the un-treated controls. In the case of KJ40E-dipped seeds, glutathione reductase was also increased compared to control. Exogenous osmotic stress has been found to induce the overproduction of reactive oxygen species, which can cause membrane lipid peroxidation [55]. To mitigate this, plants have developed defensive mechanisms for scavenging reactive oxygen species; these mechanisms involve the production of enzymatic antioxidants such as ascorbate peroxidase, catalase, glutathione peroxidase, glutathione reductase, glutathione S-transferase, NADPH oxidase, and superoxide dismutase, as well as non-enzymatic antioxidants, including ascorbate, glutathione, and phenolic compounds [56]. Sheteiwy et al. [57] demonstrated that GABA-treated rice seeds exhibit increased germination percentage under osmotic stress conditions, with MDA and hydroperoxide decreasing in activity, while that of other antioxidant enzymes increased. Similarly, melatonin-soaked soybean seeds showed an increased germination rate under 6% PEG conditions, with a reduction in MDA, hydroperoxide, and superoxide anion levels; and a concomitant increase in the production of antioxidant enzymes, including catalase, peroxidase, and ascorbate peroxidase [58]. Therefore, the increase in catalase, glutathione peroxidase, and glutathione reductase activities which we also observed in our present work, indicated that H30-3E and KJ40E decreased membrane lipid peroxidation by increasing enzymatic antioxidant activities.

Breaking seed dormancy and germination are regulated by light, moisture, and temperature as well as by modulating expression of the plant hormones GA and ABA [59, 60]. The GA stimulates seed germination, plant growth, and development by promoting hydrolytic enzyme production, which induces softening of the seed cover [61]. Gibberellins are synthesized through the terpenoid pathway: within plastids, geranylgeranyl diphosphate is transformed to *ent*-kaurene through the actions of *ent*-copalyl diphosphate synthase and *ent*-kaurene synthase. Following this, *ent*-kaurene is further processed into GA12 through *ent*-kaurene oxidase and *ent*-kaurenoic acid oxidase. Finally, GA12 is transformed to bioactive GA4 through the activities of the GA 20-oxidase (GA20ox) and GA 3-oxidase (GA3ox) in the cytoplasm [62]. In this pathway, *GA3* encodes for *ent*-kaurene oxidase [63] that converts GA12 within the plastids, while the respective enzymes encoded by *GA20ox1* and *GA3ox1* convert bioactive GA4 in the cytoplasm, which is fundamental for GA biosynthesis [64, 65]. Our qRT-PCR results showed that relative gene expression under drought conditions was upregulated for *GmGA3* at 6 h after treatment, and it could lead to a subsequent marked accumulation of GA content in KJ40E-dipped seeds. Increased or excessively active GA correlated with a significantly enhanced expression of *GmGA2ox1*, which is involved in the inhibition of the GA biosynthesis pathway, at 12 h after treatment [66]. The ABA is one of non-hydraulic root-sourced signals (nHRS) accumulated by plants when drought stress occurs and is known to induce tolerance by stomata closure in plants [67–69]. However, in seed, it is associated with the

maintenance of seed dormancy, and *ABI4*, *ABI5*, and *DERB1* are known to be important genes for ABA signaling. The *ABI4* and *ABI5* positively regulate ABA signaling [70, 71]. The expression of *DERB1* gene is controlled by abiotic stresses such as drought, salt concentration, cold, and heat. Additionally, *DERB1* affects the expression of abscisic acid-responsive element (ABRE) -related genes via an ABA-independent pathway [49, 72, 73]. We observed that *GmABI4* expression in H30-3E- or KJ40E-dipped seeds was upregulated at 12 h after treatment, with a similar increase also noted for *GmDREB1* at 12 h after treatment regardless of treatment. The ABA content was not affected by the treatment. The ratio of GA to ABA was significantly increased only in KJ40E-dipped seeds compared to the control, suggesting that KJ40E could be involved in the modulation of the biosynthesis of plant hormones, such as GA, during the germination stage, which can lead to a decrease in seed germination by osmotic stress.

Seeds require energy from stored sucrose and the raffinose family oligosaccharides (RFOs) during germination process [74]. Free sugars, including sucrose, stachyose, fructose, glucose, raffinose, and galactose, are reduced during germination [75] and used as an energy source. Our findings showed that the sucrose, glucose, and fructose contents tended to decrease regardless of PEG conditions and treatment; however, the time-dependent changes seemed to be different. Sucrose content drastically decreased in 20% PEG, with slope values of 0.0699 for the control, 0.0978 for H30-3E, and 0.756 for KJ40E, which were higher than 0.0278, 0.0439, and 0.0236, respectively, in the 0% PEG reference condition. The glucose content in the KJ40E-dipped seeds, and the fructose content in H30-3E-dipped seeds, also steeply decreased in 20% relative to 0% PEG. These results suggest that osmotic stress tended to promote the rapid degradation of sucrose to glucose and fructose, and that glucose and fructose could be effectively used as energy sources in the germination stage in KJ40E- or H30-3E-dipped seeds, which could be associated with the increased gemination rates observed in the extract-dipped seeds under 20% PEG conditions.

In conclusion, our study's findings suggest that the bacterial extracts of H30-3E and KJ40E could indirectly alleviate the decrease in soybean germination under osmotic stress caused by 20% PEG treatment. This alleviation is likely achieved through the regulation of secondary oxidative stress responses, accomplished by enhancing the activities of antioxidant enzymes (CAT, GPX, and GR). In terms of gene expression, KJ40E-treated seeds exhibited regulated GA biosynthesis genes and ABA response genes, whereas H30-3E-treated seeds displayed upregulated ABA response genes. Consequently, KJ40E-treated seeds showed a significant accumulation of GA, leading to an increased GA-to-ABA ratio. In both H30-3E and KJ40E-treated seeds, glucose and fructose were likely utilized as energy sources to adapt to osmotic conditions. Overall, KJ40E and H30-3E had the potential for enhancing tolerance to osmotic stress during soybean seed germination by inducing physiological changes in the seeds, which could ultimately contribute to improved crop yield and quality under challenging environmental conditions.

## Supporting information

**S1 Fig. Selection of bioactive extracts.** Final germination percentage of KJ40 (A), H30-3 (B), and H26-2 (C) under 0% or 20% PEG. Data presented as means + standard error; small letters on the bar mean significant difference (n=6, statistical significance assessed by the LSD test). (TIF)

**S1 Table. One-way anova analysis of the final germination percentage between conditions.** (DOCX)

## Author Contributions

**Conceptualization:** Mee Kyung Sang.

**Formal analysis:** Sang Tae Kim.

**Investigation:** Sang Tae Kim.

**Methodology:** Sang Tae Kim.

**Supervision:** Mee Kyung Sang.

**Visualization:** Sang Tae Kim.

**Writing – original draft:** Sang Tae Kim.

**Writing – review & editing:** Mee Kyung Sang.

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
