## [Decision Letter · Decision Letter 0]

25 Jul 2023

PONE-D-23-14276Enhancement of osmotic stress tolerance in soybean seed germination by bacterial bioactive extractsPLOS ONE

Dear Dr. SANG,

Thank you for submitting your manuscript to PLOS ONE. After careful consideration, we feel that it has merit but does not fully meet PLOS ONE’s publication criteria as it currently stands. Therefore, we invite you to submit a revised version of the manuscript that addresses the points raised during the review process.

We look forward to receiving your revised manuscript.

Kind regards,

Meenakshi Thakur, Ph.D.

Academic Editor

PLOS ONE

Journal Requirements:

a) The name of the colleague or the details of the professional service that edited your manuscript.

b) A copy of your manuscript showing your changes by either highlighting them or using track changes (uploaded as a *supporting information* file).

c) A clean copy of the edited manuscript (uploaded as the new *manuscript* file).

"This work was supported by National Institute of Agricultural Sciences (Project no. PJ01587101) of Rural Development Administration, Republic of Korea."

**Additional Editor Comments:**

The research article has been written well by the authors. However, some points are there which must be cleared. The Introduction part should be framed in respect of the current research's objectives. Material and Methods should be complete. Results part should be more qualitative. Conclusions should be clear. The overall English grammar of the manuscript needs to be improved. The manuscript can be accepted after these changes and those suggested by the reviewers.

Reviewers' comments:

Reviewer's Responses to Questions

**Comments to the Author**

1. Is the manuscript technically sound, and do the data support the conclusions?

Reviewer #1: Yes

Reviewer #2: Yes

Reviewer #3: Partly

2. Has the statistical analysis been performed appropriately and rigorously? 

Reviewer #1: Yes

Reviewer #2: Yes

Reviewer #3: Yes

3. Have the authors made all data underlying the findings in their manuscript fully available?

Reviewer #1: Yes

Reviewer #2: Yes

Reviewer #3: No

4. Is the manuscript presented in an intelligible fashion and written in standard English?

Reviewer #1: Yes

Reviewer #2: Yes

Reviewer #3: No

5. Review Comments to the Author

Reviewer #1: The current study entitled as “Enhancement of osmotic stress tolerance in soybean seed germination by bacterial bioactive extracts” evaluate the bioactive extracts of two biostimulant bacterial strains, Bacillus butanolivorans KJ40 and B. siamensis H30-3, for their ability to convey tolerance to osmotic stress in soybean seeds during germination. This seed treatment could be a promising potential in agricultural applications aimed at protecting crops from periods of drought, thereby increasing crop yield and quality.

I believe that this study offers new findings and is therefore worthy of publication after minor revision. My suggestions are as follows.

Abstract

Line 46-48: The future perspective is not very much clear, I suggest to rewrite the concluding remark sentence and “may show” should be replaced.

Introduction

Line 54: This study is not country specific so I recommend please remove “including the United States, Brazil, Argentina, and China”.

Line 63: please remove and here in the sentence “physiological and biochemical”, and it should be replaced with coma (,).

Line 66-67: Water stress and drought have same meaning and only one word should be used here. Please re write this sentence as follows “Soybeans are highly sensitive to water stress, and it can severely impact their growth and production”.

Line 70-72: “During the plant’s maturation, drought stress can also cause wilting, leaf yellowing, and premature leaf drop owing to increased water loss through transpiration in response to elevated temperatures” should be rewritten as During the plant’s maturation, drought stress also cause wilting, leaf yellowing, and premature leaf drop owing to increased water loss through transpiration in response to elevated temperatures”.

I suggest in Introduction section do not use so much can, since Introduction section should be written in the present tense, either in active voice form or passive voice form so I suggest author must consider the English grammar rules.

Materials and methods

Line 129-131: “Glycine max ‘Daewon’ was surface-sterilized by in 2% sodium hypochlorite,” Remove by in this sentence.

Results

Some sentences are very long and I suggest need to break it into shorter ones.

Discussion

I recommend to cite these references about ABA and drought stress in the Discussion section such as:

Batool et al., 2019; Partial and full root-zone drought stresses account for differentiate root-sourced signal and yield formation in primitive wheat”. Plant Methods

Gui et al., 2020; Differentiate effects of non-hydraulic and hydraulic root signaling on yield and water use efficiency in diploid and tetraploid wheat under drought stress”. Environmental and Experimental Botany

Lv et al., 2019; Comparative response to drought in primitive and modern wheat: a cue on domestication. Planta

Line 340: “with a decrease in oxidative stress Our findings indicated that MDA content was higher in seeds” Please add a full stop here between the two sentences.

Line 367-368: “The GA stimulates seed germination, plant growth, and development by promoting hydrolytic enzyme production and inducing softening of the seed cover [54]”. And should be replaced with by i.e. “production and inducing”.

Conclusion

Please elaborate Conclusion section more precisely.

Overall, the paper is well-explained, and the efforts of the Authors are praiseworthy in this regard. However, I recommend, thoroughly improve the writing of the paper in the context of English grammar and sentence structure.

Reviewer #2: This manuscript reported the study on physiological and gene expression changes of the soybean seeds, challenged by two biostimulant bacterial strains KJ40 and H30-3. The authors found that the dip-treatment of soybean seeds can enhance resistance to osmotic stress during germination, which shows promising potential in agricultural application in increasing soybean yield and quality. This manuscript is well-organized, well-written, with high quality figures.

Minor comments:

1. The short term HAT is not appropriate in many places, such as Abstract, figures. It is better just use the full-name with only three words "hours after treatment".

2. line 174, "as described by xx", should be the author.

3. page 9, what is the meaning for "two experiments", it might be better as "two treatments"?

4. In Table 2, a and b in the footnote is the same as a,b,c for statistic test, so it is suggested to change one of them.

5. In the Results part form lines 254 to line 285, it is better to divided it into two to three paragraph, to clearly show the GA and ABA pathway genes.

6. Styles of some references should be carefully checked.

Reviewer #3: The manuscript entitled “Enhancement of osmotic stress tolerance in soybean seed germination by bacterial bioactive extracts” by Sang Tae Kim and Mee Kyung Sang tackles an important economic and societal issue. The increasing world population and the demand to decrease the amount of agrochemicals cause new biological solutions for agriculture are of crucial importance. Although the manuscript tackles an important question the novelty of this manuscript is rather limited. The authors describe the impact of bio-extracts from two plant-growth-promoting bacteria on soybean germination, basic stress-related parameters, and the expression of genes encoding enzymes responsible for ABA and GA synthesis. The manuscript is descriptive and does not provide any novel insight into the mechanisms underlying bacteria-plant interactions. The analysed parameters i.e. MDA, GSH, antioxidant enzymes, ABA, etc. are typical parameters analysed in this kind of research and in fact, the authors did not provide any novel conclusion based on their results. The authors neither analyse the nature of compounds present in obtained extracts nor provide any hypothesis on the nature of these compounds. This manuscript is one of several other very similar manuscripts showing the potential beneficial effect of one/few bacteria on one/few plant/s without providing any general conclusions.

The introduction is too generic. It is not clear from the introduction why these parameters and genes were analysed. Several pieces of information which should be in the introduction are given in the discussion, for example lines 365-376, lines 381-386. Moreover, the discussion contains a repetition of results in too many details. Therefore, the discussion is too descriptive and the results are not properly discussed.

Materials and methods are quite well described however some details are missing.

Lines 121-123 – it is not clear how the extraction was performed.

Line 130 – it is not clear how much of the extract was used since 100 μg/ml is not informative. What is 100 μg?

Line 135 – it is unclear what the authors meant by fraction and concentration selection.

Line 136 – what light conditions were used?

The description of results should be more qualitative.

6. PLOS authors have the option to publish the peer review history of their article (what does this mean?). If published, this will include your full peer review and any attached files.

Reviewer #1: **Yes: **Professor Dr You-Cai Xiong

Reviewer #2: No

Reviewer #3: No

---

## [Author Response · Author response to Decision Letter 0]

28 Aug 2023

Reviewer #1

Line 46-48: The future perspective is not very much clear, I suggest to rewrite the concluding remark sentence and “may show” should be replaced.

Response: We have made a change to the sentence (lines 47-48)

Line 54: This study is not country specific so I recommend please remove “including the United States, Brazil, Argentina, and China”.

Response: We have removed the phrase “including the United States, Brazil, Argentina, and China” (line 54)

Line 63: please remove and here in the sentence “physiological and biochemical”, and it should be replaced with coma (,).

Response: We have replaced the word “and” with a coma (,) (line 62)

Line 66-67: Water stress and drought have same meaning and only one word should be used here. Please re write this sentence as follows “Soybeans are highly sensitive to water stress, and it can severely impact their growth and production”.

Response: Thank you for the comment. We have made a change to the sentence (line 65)

Line 70-72: “During the plant’s maturation, drought stress can also cause wilting, leaf yellowing, and premature leaf drop owing to increased water loss through transpiration in response to elevated temperatures” should be rewritten as During the plant’s maturation, drought stress also cause wilting, leaf yellowing, and premature leaf drop owing to increased water loss through transpiration in response to elevated temperatures”.

Response: Thank you for your feedback. We have made the change to the sentence as you suggested. (lines 77-80)

Line 129-131: “Glycine max ‘Daewon’ was surface-sterilized by in 2% sodium hypochlorite,” Remove by in this sentence.

Response: We removed the word “by” from the sentence (line 141)

Some sentences are very long and I suggest need to break it into shorter ones.

Response: We restructured several sentences as follows:

∙ Lines 198-215 have been divided into lines 211-220 and 221-229

∙ Lines 223-239 have been split into lines 237-248 and 249-256

∙ Lines 254-285 have been reorganized into lines 271-293 and 294-305 

I recommend to cite these references about ABA and drought stress in the Discussion section such as:

Batool et al., 2019; Partial and full root-zone drought stresses account for differentiate root-sourced signal and yield formation in primitive wheat”. Plant Methods

Gui et al., 2020; Differentiate effects of non-hydraulic and hydraulic root signaling on yield and water use efficiency in diploid and tetraploid wheat under drought stress”. Environmental and Experimental Botany

Lv et al., 2019; Comparative response to drought in primitive and modern wheat: a cue on domestication. Planta

Response: We have incorporated an additional sentence along with the provided references in the discussion section (lines 400-402)

Line 340: “with a decrease in oxidative stress Our findings indicated that MDA content was higher in seeds” Please add a full stop here between the two sentences.

Response: We have separated the two sentences (lines 359-360)

Line 367-368: “The GA stimulates seed germination, plant growth, and development by promoting hydrolytic enzyme production and inducing softening of the seed cover [54]”. And should be replaced with by i.e. “production and inducing”.

Response: We have replaced the sentence (lines 387)

Please elaborate Conclusion section more precisely.

Response: We have made change the paragraph (line 428-440)

Reviewer #2

1. The short term HAT is not appropriate in many places, such as Abstract, figures. It is better just use the full-name with only three words "hours after treatment".

Response: We have replaced the words (line 39, 44, 239-243, 245-247 250-253, 273-274, 276-284, 286-287, 289, 291-292, 294-297, 299-300, 302, 327, 397, 400, and 408-409. figure 1, 2, 3, 4, and 6)

2. line 174, "as described by xx", should be the author.

Response: We have added author’s name in place of “xx” (line 187)

3. page 9, what is the meaning for "two experiments", it might be better as "two treatments"?

Response: We pooled the data from two sets of experiments based on a statistical analysis of the homogeneity of variance.

4. In Table 2, a and b in the footnote is the same as a,b,c for statistic test, so it is suggested to change one of them.

Response: We have made a change in the footnote (Table 2)

5. In the Results part form lines 254 to line 285, it is better to divided it into two to three paragraph, to clearly show the GA and ABA pathway genes.

Response: We have divided two paragraph (lines 271-293 and 294-305)

6. Styles of some references should be carefully checked.

Response: We have checked the references

Reviewer #3

The introduction is too generic. It is not clear from the introduction why these parameters and genes were analysed. Several pieces of information which should be in the introduction are given in the discussion, for example lines 365-376, lines 381-386. Moreover, the discussion contains a repetition of results in too many details. Therefore, the discussion is too descriptive and the results are not properly discussed.

Response: We have added some sentences (lines 69-77)

Lines 121-123 – it is not clear how the extraction was performed.

Response: We have replaced the extraction methods (lines 128-132)

Line 130 – it is not clear how much of the extract was used since 100 μg/ml is not informative. What is 100 μg?

Response: We have added the amount of extraction (lines 133-137)

Line 135 – it is unclear what the authors meant by fraction and concentration selection.

Response: We have replaced the unclear phrase “fraction and concentration selection” with a more precise explanation (lines 147-149)

Line 136 – what light conditions were used?

Response: We have added light conditions (line 149)

The description of results should be more qualitative.

Response: We restructured several sentences as follows:

∙ Lines 198-215 have been divided into lines 211-220 and 221-229

∙ Lines 223-239 have been split into lines 237-248 and 249-256

∙ Lines 254-285 have been reorganized into lines 271-293 and 294-305 

We have adjusted the references number due to the inclusion of additional sentences in the manuscript.

Lines 72, 74, 77, 80, 82, 85, 87-88, 92-93, 95, 97, 99, 104, 107-108, 111, 114, 127, 145, 156, 162, 167, 183, 187, 339, 351, 360, 369, 374, 380, 385, 387, 393, 395, 400, 402, 404, 407, and 415-416; Table 1

---

## [Editor Report · Decision Letter 1]

2 Oct 2023

Enhancement of osmotic stress tolerance in soybean seed germination by bacterial bioactive extracts

PONE-D-23-14276R1

Dear Dr. Sang,

We’re pleased to inform you that your manuscript has been judged scientifically suitable for publication and will be formally accepted for publication once it meets all outstanding technical requirements.

Kind regards,

Meenakshi Thakur, Ph.D.

Academic Editor

PLOS ONE

Additional Editor Comments (optional):

The manuscript is accepted for publication in the present form.
---

## [Editor Report · Acceptance letter]

4 Oct 2023

PONE-D-23-14276R1 

Enhancement of osmotic stress tolerance in soybean seed germination by bacterial bioactive extracts 

Dear Dr. Sang:

I'm pleased to inform you that your manuscript has been deemed suitable for publication in PLOS ONE. Congratulations! Your manuscript is now with our production department. 

Kind regards, 

on behalf of

Dr. Meenakshi Thakur 

Academic Editor

PLOS ONE